# Machine-Learning-Based Prediction Modeling for Debris Flow Occurrence: A Meta-Analysis

Lianbing Yang [1,2,3], Yonggang Ge [1,2,*], Baili Chen [3,4], Yuhong Wu [2,3] and Runde Fu [3,5]

1   Key Laboratory of Mountain Hazards and Earth Surface Process, Chinese Academy of Sciences, Chengdu 610299, China; ylb@imde.ac.cn
2   Institute of Mountain Hazards and Environment, Chinese Academy of Sciences, Chengdu 610299, China; wyh@imde.ac.cn
3   University of Chinese Academy of Sciences, Beijing 101408, China; furunde7691@igsnrr.ac.cn (R.F.)
4   Northwest Institute of Eco-Environment and Resources, Chinese Academy of Sciences, Lanzhou 730000, China
5   Institute of Geographic Sciences and Natural Resources Research, Chinese Academy of Sciences, Beijing 100101, China
*   Correspondence: gyg@imde.ac.cn

**Abstract:** Machine learning (ML) has become increasingly popular in the prediction of debris flow occurrence, but the various ML models utilized as baseline predictors reported in previous studies are typically limited to individual case bases. A comprehensive and systematic evaluation of existing empirical evidence on the utilization of ML as baseline predictors for debris flow occurrence is lacking. To address this gap, we conducted a meta-analysis of ML-based prediction modeling of debris flow occurrence by retrieving papers that were published between 2000 and 2023 from the Scopus and Web of Science databases. The general findings were as follows: (1) A total of 84 papers, distributed across 37 different journals in this time period, reflecting an overall upward trend. (2) Debris flow disasters occur throughout the world, and a total of 13 countries carried out research on the prediction of debris flow occurrence based on ML; China made significant contributions, but more research efforts in African countries should be considered. (3) A total of 36 categories of ML models were utilized as baseline predictors for debris flow occurrence, with logistic regression (LR) and random forest (RF) emerging as the most popular choices. (4) Feature engineering and model comparison were the most commonly utilized strategies in predicting debris flow occurrence based on ML (53 and 46 papers, respectively). (5) Interpretation methods were rarely utilized in predicting debris flow occurrence based on ML, with only 16 papers reporting their utilization. (6) In the prediction of debris flow occurrence based on ML, interpretation methods were rarely utilized, searching by data materials was the most important sample data source, the topographic factors were the most commonly utilized category of candidate variables, and the area under the ROC curve (AUROC) was the most frequently reported evaluation metric. (7) LR's prediction performance for debris flow occurrence was inferior to that of RF, BPNN, and SVM; SVM was comparable to RF, and all superior to BPNN. (8) The application process for the prediction of debris flow occurrence based on ML consisted of three main steps: data preparation, model construction and evaluation, and prediction outcomes. The research gaps in predicting debris flow occurrence based on ML include utilizing new ML techniques and enhancing the interpretability of ML. Consequently, this study contributes both to academic ML research and to practical applications in the prediction of debris flow occurrence.

**Keywords:** machine learning; debris flow; occurrence; prediction; meta-analysis



## 1. Introduction

Debris flow is a frequent natural geological phenomenon in valleys, which is a three-phase saturated fluid composed of solids, liquids, and gases [1,2]. Its formation is catalyzed by triggering conditions such as heavy rains, glacial and snowmelt waters, and dam

failures [3,4]. In recent years, the occurrence of debris flow disasters has risen due to extreme weather, earthquakes, forest fires, and human engineering activities [5]. Debris flow is characterized by sudden and rapid movement, which can quickly erode, transport, accumulate, and impact the earth's surface [6,7]. This phenomenon has posed a serious threat to human life and property, and the ecological environment of mountainous regions, and therefore, it has become a major disaster factor hindering the social and economic development of mountain areas around the world [8,9]. Proactive deployment of disaster prevention and mitigation measures based on predictive information regarding debris flow occurrence can minimize the impact of such disasters [10–12]. Hence, the accurate and scientific prediction of debris flow occurrence holds immense significance in effectively preventing and managing debris flow disasters.

The genesis of debris flow is fundamentally governed by three primary factors: material source, water source, and topography [13–15]. Moreover, the causative factors behind debris flow are intricate, involving a multitude of elements such as geology, topography, landform, soil, vegetation, rainfall, and temperature [16–20]. Moreover, the formation process of debris flow involves theoretical knowledge of many disciplines, and shows complex nonlinear characteristics [21]. Consequently, numerous researchers have endeavored to construct qualitative or quantitative models to comprehensively simulate the intricate mechanisms underlying debris flow formation, thereby enhancing the accuracy of prediction of debris flow occurrence [22–26]. In this study, the prediction of debris flow occurrence refers to whether debris flow occurs or the possibility of debris flow occurrence within a certain area based on the prediction model. The possibility of debris flow occurrence is generally characterized by the susceptibility [27,28] or hazard [29,30] level of the debris flow. Susceptibility refers to the possibility of debris flow occurrence in a certain evaluation unit, considering non-triggering factors such as topography, geomorphology, and surface cover characteristics. Hazard, on the other hand, incorporates triggering factors like rainfall into the susceptibility assessment. Debris flow prediction models fall into four main categories: knowledge-driven models (analytic hierarchy process [31,32], rainfall threshold method [33–35], geomorphic information entropy [36], etc.), traditional statistical models (weight of evidence method [37,38], certainty factor [39,40], frequency ratio [40,41], etc.), numerical simulation models (FLO-2D [42], Flow-3D [43,44], Debris2D [45], etc.), and ML models (LR [46,47], RF [24,48], convolutional neural networks [49,50], etc.).

ML employs specific algorithms within computers to discern patterns within data and construct models [51,52]. Owing to its rapid advancement in recent years and its robust capability to capture intricate relationships between predictors and response variables, ML has gained substantial traction in predicting debris flow occurrence [22,53,54]. While numerous studies have affirmed the effectiveness, applicability, and advantages of utilizing ML models as baseline predictors for debris flow occurrence on an individual case basis, these singular instances provide limited reference information. Therefore, there is a need to systematically summarize the research findings related to the utilization of ML models as baseline predictors for debris flow occurrence. To address these issues, this study collated journal papers published between 2000 and 2023 from the Scopus and Web of Science databases to consolidate the collective knowledge concerning the prediction of debris flow occurrence based on ML. Subsequently, a meta-analysis was conducted within this domain.

## 2. Data Processing Workflow

### 2.1. Literature Retrieval and Selection Criteria

To comprehensively retrieve literature highly relevant to the research topic, while reducing the subsequent manual screening workload, this study utilized a three-level search approach, incorporating conditions related to the problem, model, and other factors, and linking them using the logical operator "and" (Figure 1). At the problem level, the article's title was required to incorporate words characterizing debris flow and terms associated with the prediction of debris flow occurrence. For this purpose, synonymous terms for debris flow, such as debris slide, debris flood, and mudflow, were considered. Additionally, words

related to debris flow prediction, including occurrence, initiation, prediction, assessment, warning, and modeling, were also included in the search criteria. At the modeling level, the abstract was mandated to include terms related to ML, considering aspects like commonly used ML models and terminologies. At the other level, we included English-language papers published between 2000 and 2023 (as there were fewer studies before 2000); we excluded review articles and conference papers.

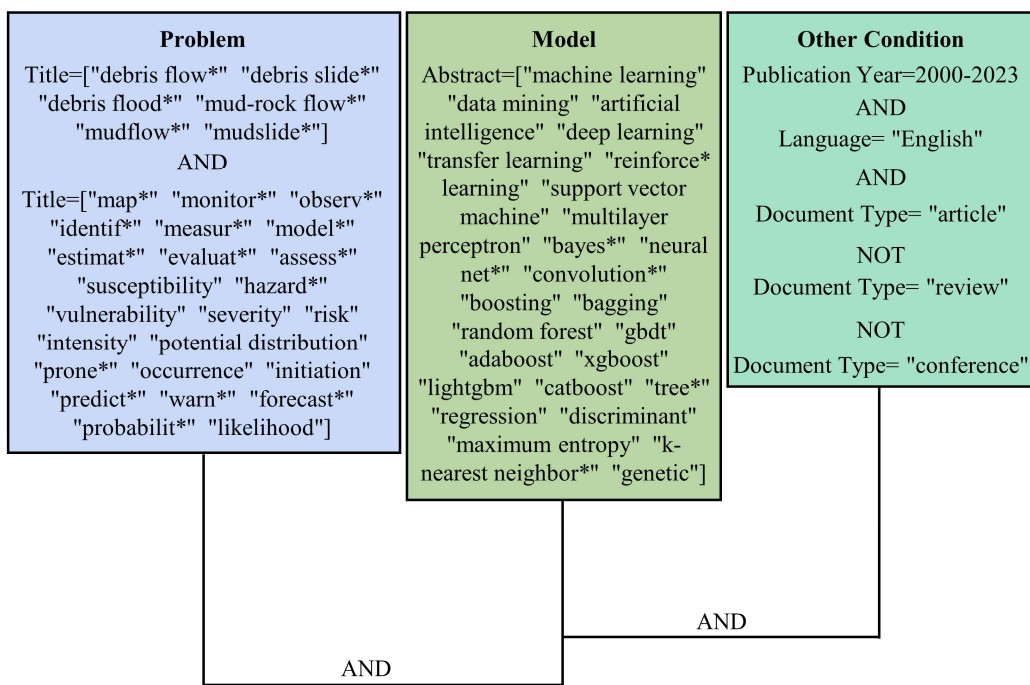

**Figure 1.** Article search query design. The logical relationship of the words in the square brackets is OR.

Scopus and Web of Science, renowned literature databases covering diverse fields, were utilized for the literature search. Based on the criteria outlined in Figure 1, this study systematically searched for literature related to ML-based prediction of debris flow occurrence in the Scopus and Web of Science databases on 27 December 2023. Afterward, following the Preferred Reporting Items for Systematic Reviews and Meta-Analyses methodology (PRISMA) [55], papers for inclusion in this study were selected. The PRISMA framework comprises four key phases: "identification", "screening", "eligibility", and "included". In addition, in this study, the PRISMA selection criteria primarily encompassed two aspects: (1) inclusion of papers featuring one or more ML models as a baseline predictor for debris flow occurrence, with a clear training and verification process, and quantitative prediction performance evaluation; (2) exclusion of papers focused on purely experimental methods for predicting debris flow occurrence.

Firstly, in the "identification" phase, we retrieved 410 journal papers from the two databases and removed 189 duplicate papers. Secondly, during the "screening" phase, we assessed the titles and abstracts of 221 papers, eliminating 64 papers that did not meet the PRISMA selection criteria. Thirdly, in the "eligibility" phase, we scrutinized the 157 remaining papers in detail, and after eligibility assessment, we identified 73 papers as unrelated to this meta-analysis and removed them. Finally, in the "included" phase, 84 papers were deemed suitable for inclusion in this meta-analysis (Figure 2).

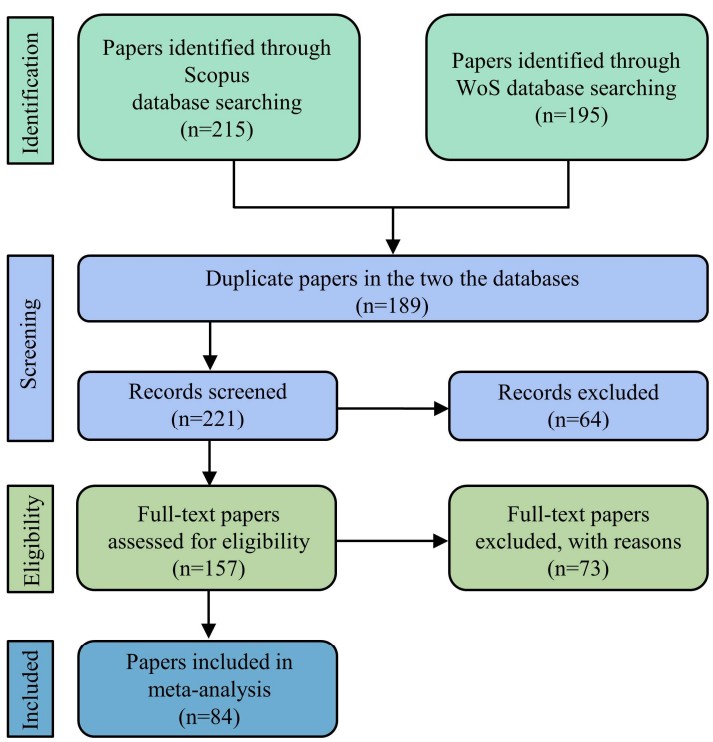

**Figure 2.** PRISMA flowchart demonstrating the selection of papers.

### 2.2. Data Extraction

The data used for the meta-analysis in this study were collected from the 84 papers, and the results extracted from each paper were compiled into a structured database. As outlined in Table 1, the database encompassed 15 attribute fields derived from the fundamental aspects of papers, and the modeling and prediction of debris flow occurrence using ML. The attribute fields included journal, title, year, study area, institution, type of occurrence, mapping unit, baseline model, improvement strategy, sample data, candidate variable, validation technique, evaluation metric, area under the curve, and case number. The specific meanings of each attribute field are shown in the attribute description column in Table 1.

**Table 1.** Database attribute fields for meta-analysis.

| ID | Field Name | Description | Type |
|----|-----------|-------------|------|
| 1 | Journal | Name of journal | Text |
| 2 | Title | Title of paper | Text |
| 3 | Year | Year of publication | Numeric |
| 4 | Study area | Country of study area of paper | Text |
| 5 | Institution | Country of first research institution | Text |
| 6 | Type of occurrence | Examples include occurrence or nonoccurrence, susceptibility assessments, hazard assessment of debris flow | Text |
| 7 | Evaluation unit | Unit utilized for prediction of debris flow occurrence | Text |
| 8 | Baseline model | ML utilized as baseline predictor of debris flow occurrence | Text |
| 9 | Improvement strategy | Modeling strategy for improving performance of ML utilized for predictor of debris flow occurrence | Text |
| 10 | Sample data | Source of debris flow sample data | Text |
| 11 | Candidate variables | Candidate feature utilized for prediction of debris flow occurrence based on ML | Text |
| 12 | Validation technique | Method used to divide the training set and test set utilized for prediction of debris flow occurrence based on ML | Text |

**Table 1.** *Cont.*

| ID | Field Name | Description | Type |
|----|-----------|-------------|------|
| 13 | Evaluation metric | Metrics utilized to report performance of the ML model utilized for prediction of debris flow occurrence | Text |
| 14 | Area under the curve | Prediction area under the ROC curve of debris flow occurrence based on ML | Numeric |
| 15 | Number of cases | Number of combinations of training and test sets utilized for debris flow occurrence based on ML | Numeric |

The determination of the number of cases in each paper was based on the quantity of data sets used for ML training in the respective articles. For instance, if an article incorporated three training sets, all utilized for ML modeling in the prediction of debris flow occurrence with quantitative results, then the article was recorded as contributing three cases. The area under the curve was recorded on a case-by-case basis.

## 3. Results

In the research on ML-based prediction modeling of debris flow occurrence, both whether debris flow occurs or the possibility of debris flow occurrence essentially rely on ML classification models [21,28,29]. Therefore, based on the 84 papers selected in Section 2, this study performed a meta-analysis on the prediction of debris flow occurrence based on ML from an overall perspective, and did not distinguish between the types of debris flow occurrence in detail. Firstly, the general characteristics of the 84 papers were analyzed, covering the annual number of papers, the published journals, and the geographical distribution of the study areas and first institutions. Subsequently, the fundamental characteristics of the ML application in the prediction of debris flow occurrence were elaborated, considering ML categories, strategies for improving prediction performance, model interpretation, sample data sources, evaluation units, candidate variable categories, validation techniques, evaluation metrics, prediction performance, and application processes.

### 3.1. General Characteristics of Studies

During the literature search timeframe for this study (2000–2023), papers on the prediction of debris flow occurrence based on ML emerged in 2006, and continued to appear each year from 2006 to 2023, which indicated an overall upward trend (Figure 3). A total of 84 papers were published, with the lowest numbers recorded in 2008 and 2010 (1 paper each), and the highest in 2022 (18 papers). Notably, approximately 58.33% (49 out of 84) of the papers were published within the last five years (2018–2023), underscoring the increasing attention given to the prediction of debris flow occurrence based on ML in recent years.

The 84 papers were distributed across 37 journals (Table 2), with 13 journals publishing 2 or more papers. Among these, 9 publishers were involved, with significant contributions from Springer, Elsevier, and MDPI. The journal with the highest number of published papers was *Natural Hazards*, totaling 15, followed by *Remote Sensing* with 8 papers. Except for Disaster Advances, which is currently not included in the Science Citation Index (SCI), the remaining 12 journals are SCI journals, with *Engineering Geology* having the highest impact factor (7.4). In terms of subject types, these 13 journals primarily cover earth science, remote sensing science, disaster science, hydrology science, geological science, and mountain science. This diversity underscores that the prediction of debris flow involves knowledge from many subject fields.

In this study, the geographical distribution of the research areas and the first institutions of the 84 papers was determined, according to the country reported in the papers, with the exclusion of the two intercontinental-scale papers [56,57]. Out of the 84 papers, the research regions of the paper [58] corresponded to 8 countries, with the remaining papers each corresponding to a single country. The count encompassed both the country in which the research area of the paper was situated and the country in which the first institution

of the paper was located. The results are illustrated in Figure 4. The research areas of the 84 papers spanned 18 countries, distributed across 6 continents (Asia, Africa, North America, South America, Europe, and Oceania), indicating the widespread occurrence of debris flow disasters around the world. Studies on the prediction of debris flow occurrence based on ML were conducted in a total of 13 countries, showing a global interest in this field. China led in both the numbers of papers by the research areas and the first institutions, at 53 and 54, respectively, suggesting a significant contribution to this field. Notably, there was a lack of studies from African countries, with no first institution from this continent among the studies. Moreover, in the geographical distribution of the research areas, Ethiopia was the only African country reported [59]. This indicated that more study efforts on the prediction of debris flow occurrence based on ML in African countries should be considered, especially considering the severity of debris flow disasters in this region.

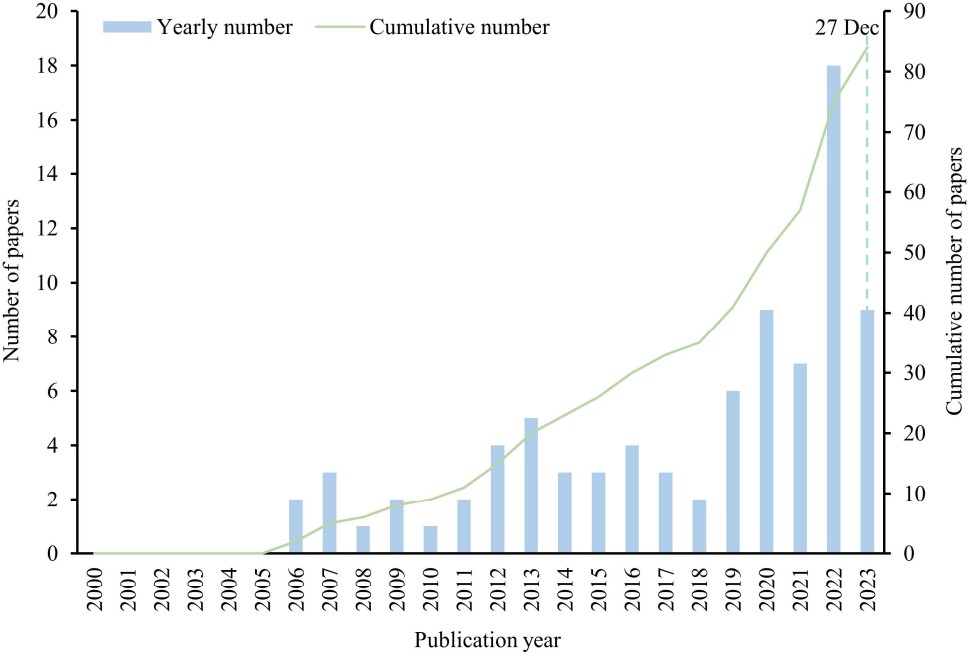

**Figure 3.** Annual and cumulative number of papers on the prediction of debris flow occurrence based on ML.

**Table 2.** Journals that published papers on the prediction of debris flow occurrence based on ML used in meta-analysis (only those journals that published 2 or more papers were included).

| Journal | Science Citation Index | Publisher | Impact Factor (2022) | Number of Papers |
|---|---|---|---|---|
| *Natural Hazards* | Yes | Springer | 3.7 | 15 |
| *Remote Sensing* | Yes | MDPI | 5.0 | 8 |
| *Engineering Geology* | Yes | Elsevier | 7.4 | 5 |
| *Water* | Yes | MDPI | 3.4 | 5 |
| *Environmental Earth Sciences* | Yes | Springer | 2.8 | 5 |
| *Natural Hazards and Earth System Sciences* | Yes | Copernicus Gesellschaft MBH | 4.6 | 4 |
| *Geomorphology* | Yes | Elsevier | 3.9 | 3 |
| *Bulletin of Engineering Geology and the Environment* | Yes | Springer | 4.2 | 3 |
| *Landslides* | Yes | Springer | 6.7 | 2 |
| *Hydrological Processes* | Yes | Wiley | 3.2 | 2 |
| *Journal of Mountain Science* | Yes | Science Press | 2.5 | 2 |
| *Open Geosciences* | Yes | De Gruyter Poland SP Z O O | 2.0 | 2 |
| *Disaster Advances* | No | Disaster Advances | None | 2 |
| *Natural Hazards and Earth System Sciences* | Yes | Copernicus Gesellschaft MBH | 4.6 | 4 |

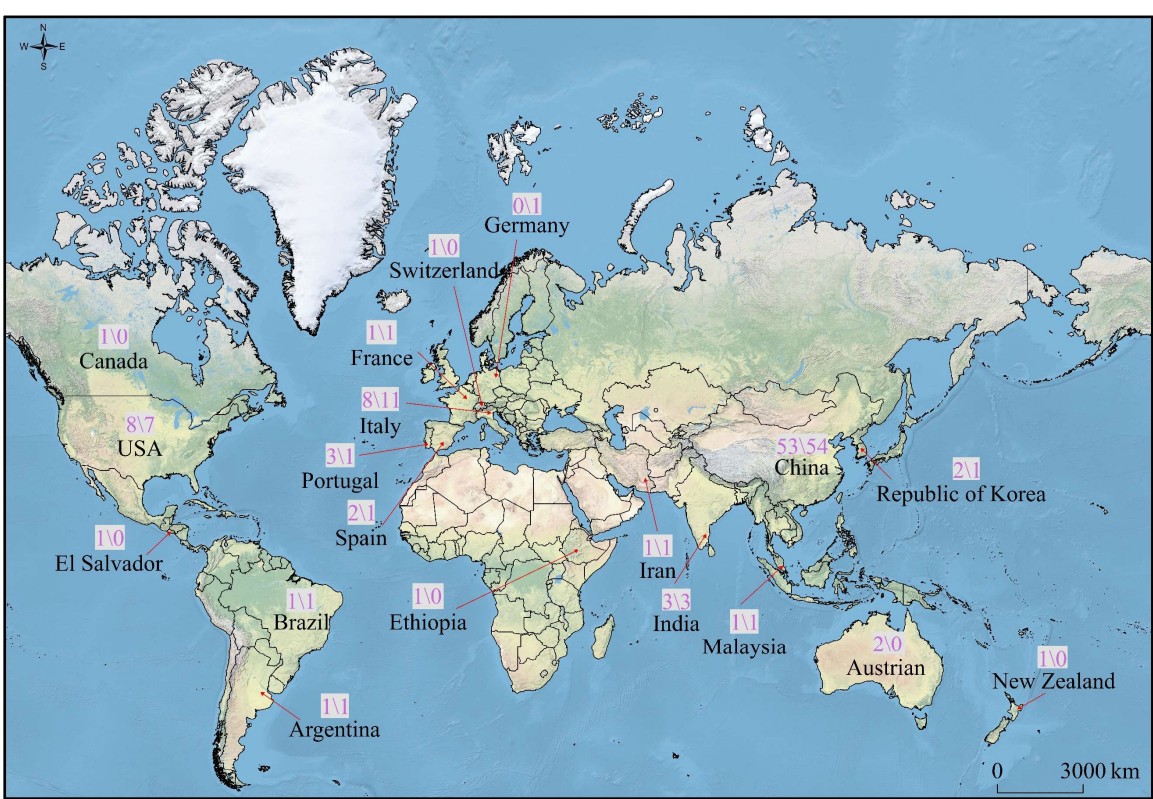

**Figure 4.** Distribution of study areas and first research institutions, according to the countries reported in the papers. The number before the slash reflects the number of papers based on the study areas; the number after slash reflects the number of papers based on first research institutions.

### 3.2. General Characteristics of ML Applications

#### 3.2.1. ML Categories

Since various ML models have been utilized as baseline predictors of debris flow occurrence in different years, relying solely on the number of papers may lead to significant errors in judging the popularity or application rate of a particular ML model in predicting debris flow occurrence. To address this, this study adopted the concept of mean value and calculated the relative annual number of papers reported for a specific ML model. The calculation formula is as follows:

$$\mu = \frac{F}{Y_{end} - Y_{begin} + 1} \tag{1}$$

In the formula, $\mu$ is the relative annual number of papers reported, $F$ is the number of papers reported for a specific ML model in the literature search period, $Y_{begin}$ is the initial year of application of a specific ML model in a certain field, and $Y_{end}$ is the last year of the literature search period.

Based on analysis of the 84 papers, the ML models utilized as baseline predictors of debris flow occurrence were summarized (Figure 5). A total of 36 categories of ML models were utilized as baseline predictors of debris flow occurrence, which were further grouped into 10 broad categories: generalized linear models, ensemble models, shallow neural networks, discriminant analysis, tree models, kernel models, Bayesian models, evolutionary models, instance-based models, and deep learning. It is worth noting that compared with other broad categories, within deep learning, only convolutional neural network was utilized as a baseline predictor of debris flow occurrence, in only five studies. This underscores the need for further study on the applicability of complex network structures in deep learning for predicting debris flow occurrence [50,60].

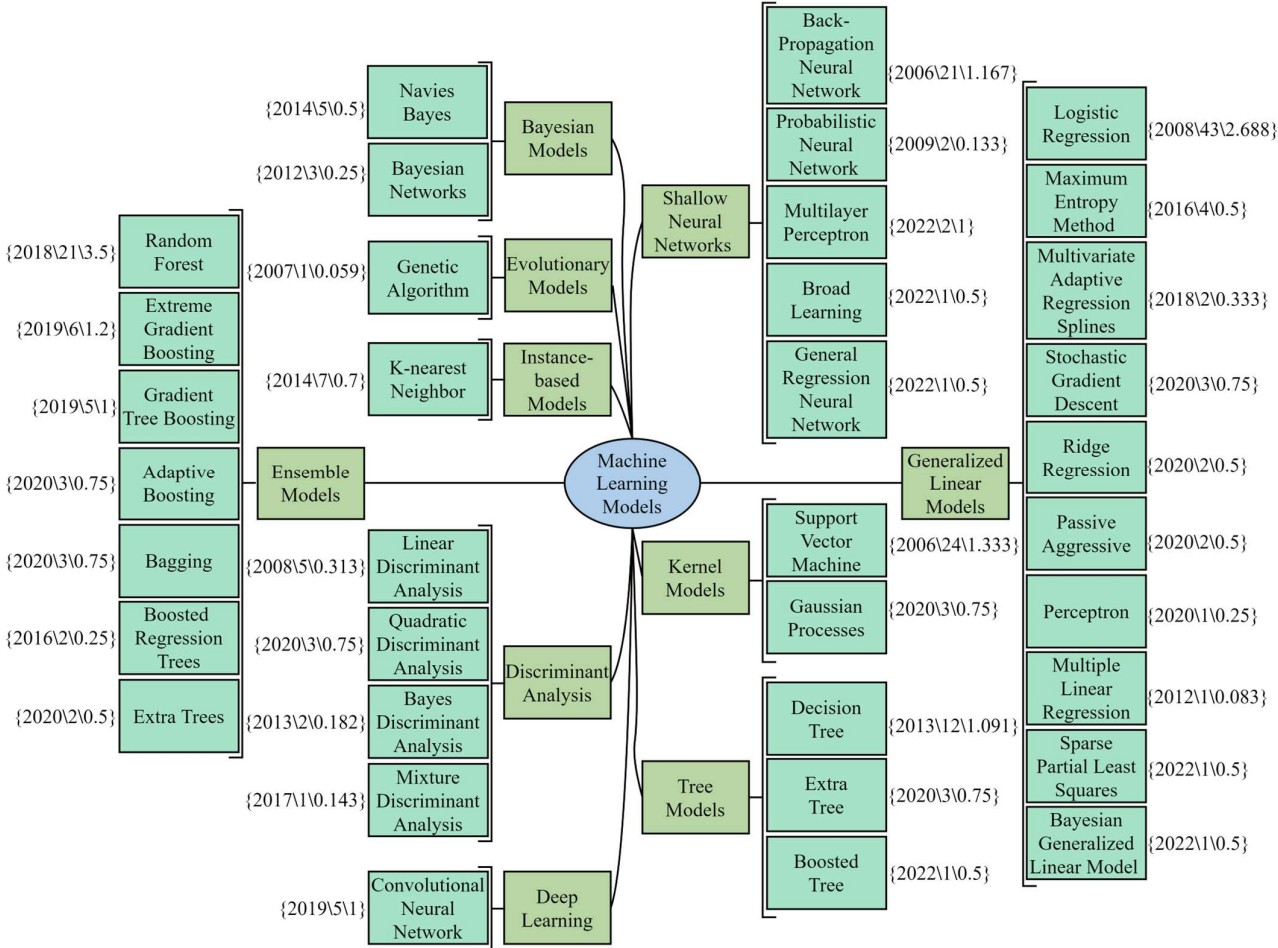

**Figure 5.** Broad categories of ML models, categories of ML models, the initial year of utilizing each category of ML model as baseline predictor, number of papers of ML for each category of ML model, and relative annual number of papers reported for each category of ML model. The three numbers in curly braces indicate the initial year, number of papers, and relative annual number of papers, respectively.

Examining the initial years of utilization for the 36 categories of ML models, BPNN (2006) and SVM (2006) were the first ML models utilized in the prediction of debris flow occurrence, followed by genetic algorithm (2007), LR (2008), linear discriminant analysis (2008), and so on. In the recent 5 years (2018–2023), 19 categories of ML models were utilized as baseline predictors of debris flow occurrence, accounting for 52.778% of the total number of ML categories retrieved in this study (2000–2023). This suggested a growing trend of utilizing a diverse range of ML models in recent years, possibly influenced by the rapid development and popularization of ML technologies [61,62].

Among the 36 categories of ML models, 4 categories were reported in more than 20 papers, in the following order: LR (43), SVM (24), BPNN (21), and RF (21). Additionally, nine categories had a relative annual number of papers greater than or equal to one, in the following order: RF (3.5), LR (2.688), SVM (1.333), extreme gradient boosting (1.2), BPNN (1.167), decision tree (1.091), gradient tree boosting (1), convolutional neural network (1), and multilayer perceptron (1). Among these, extreme gradient boosting, gradient tree boosting, convolutional neural network, and multilayer perceptron were the most utilized ML models in the prediction of debris flow in the recent 5 years, indicating their strong popularity in predictive modeling of debris flow occurrence. Considering the number of papers and the relative annual number of papers for the 36 categories of ML models, LR and RF emerged as the most popular models for predicting debris flow occurrence based on ML.

### 3.2.2. Prediction Performance Improvement Strategies

In the study of prediction modeling of debris flow occurrence based on ML, researchers have employed various performance improvement strategies to obtain a better model. This study categorized the prediction performance improvement strategies utilized in the 84 studies into five groups: feature engineering, model comparison, hyperparameter tuning, model coupling, and structure optimization. Feature engineering encompasses methods such as feature selection, dimensionality reduction, and weighting. Within feature selection, the methods include stepwise feature screening, multicollinearity analysis, and importance analysis. Model comparison involves comparisons of different ML models or comparisons between ML models and non-ML models. Hyperparameter tuning focuses on optimizing ML hyperparameters using various optimization algorithms. Model coupling involves integrating different ML models or combining ML models with non-ML models. Structure optimization includes enhancing the network structure of deep learning models.

Out of the 84 studies, the number of studies using each of the five categories of ML prediction performance improvement strategies was statistically analyzed, with the results presented in Figure 6. Feature engineering and model comparison were the most commonly utilized strategies in predicting debris flow occurrence based on ML, with 53 and 46 studies, respectively. Out of the 84 studies, 72 used one or more of these strategies, constituting 85.71% of all studies, highlighting the widespread utilization of these strategies in predicting debris flow occurrence based on ML. Among these 72 studies, 36 used only one strategy, 30 used two strategies, 5 used three strategies, and 1 reported four strategies. The statistical results revealed that the most common method of model improvement in predicting debris flow occurrence based on ML involved the utilization of one or two of these strategies, while fewer studies utilized three or four of these strategies.

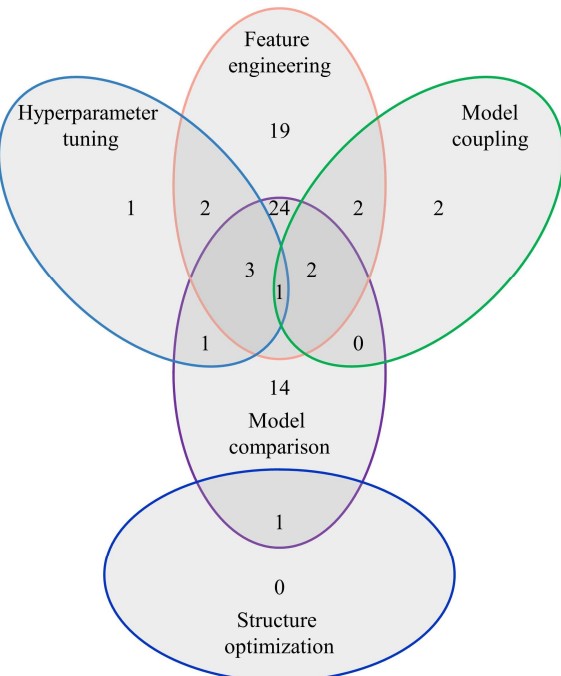

**Figure 6.** Combination of ML prediction performance improvement strategies. Numbers indicate the number of studies that utilized each strategy.

From the 53 studies utilizing feature engineering, the number of studies using each method of feature engineering was statistically analyzed. The feature selection methods within feature engineering were further subdivided, with the results presented in Figure 7. Feature selection was the most popular method of feature engineering in the prediction modeling of debris flow occurrence based on ML, with 49 studies. Among these 49 studies, the number of studies utilizing each method of feature selection was as follows: multicollinearity

analysis (26), stepwise feature screening (18), and importance analysis (13). Single methods were the most commonly utilized, rather than combinations of multiple methods for feature selection in predicting debris flow occurrence based on ML. Only a few studies used a combination of multiple methods such as multicollinearity analysis and stepwise feature screening (four) and multicollinearity analysis and importance analysis (four).

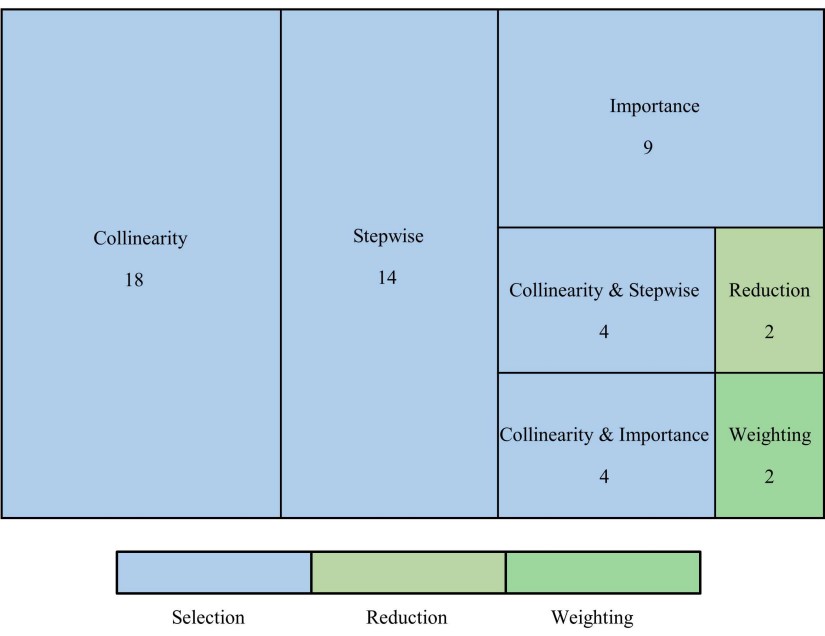

**Figure 7.** The number of studies using each feature engineering method (feature selection methods were further subdivided).

Among the 26 studies utilizing multicollinearity analysis, the used algorithms included Pearson correlation analysis [41], Spearman correlation analysis [63], variance inflation factor [64], and the tolerance method [65]. Among the 18 studies using stepwise feature screening, the methods mainly included forward selection [28], back selection [66], and artificial stepwise feature combination [67]. Among the 13 studies using importance analysis, the algorithms used included Pearson correlation analysis [68] and information gain ratio [69]. In the two studies using feature dimensionality reduction, the algorithms utilized were principal component analysis [70,71]. Among the two papers using feature weighting, the algorithms utilized were certainty factor and genetic algorithm.

Among the 46 studies using model comparison, the number of papers comparing ML models (36) was significantly higher than that of those comparing ML models and non-ML models (10), as shown in Figure 8. Among the 36 papers comparing ML models, the number of papers comparing shallow ML models (32) was much higher than that of those comparing shallow ML models and deep learning models (4). This discrepancy may be attributed to the fact that deep learning (2019) emerged much later than shallow ML (2006), and convolutional neural network was the only deep learning model utilized as a baseline predictor in the prediction of debris flow occurrence based on ML.

Among the eight studies using hyperparameter tuning, some studies utilized multiple optimization algorithms; the number of studies using each of the different optimization algorithms was as follows: grid search algorithm (four), particle swarm optimization algorithm (three), genetic algorithm (two), cuckoo optimization algorithm (one) and gray wolf optimization algorithm (one). Among the eight studies using model coupling, the number of studies using each of the model coupling methods was as follows: coupling of ML model and traditional statistical model (three), coupling of ML and mechanism model (two), and coupling between ML models (two). In one paper using structure optimization, the author improved the structure of the convolutional neural network according to the characteristics of debris flow [50].

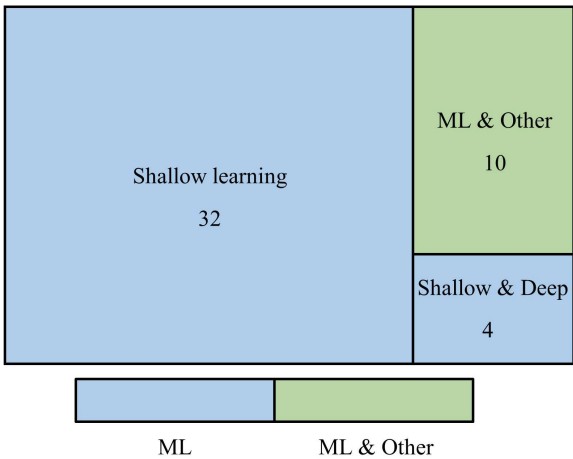

**Figure 8.** The number of studies using each type of model comparison (the comparisons between ML models were further subdivided).

### 3.2.3. Model Interpretation

ML is an inexplicable "black-box" model, so enhancing its interpretability is important to the scientific understanding of prediction outcomes of debris flow occurrence based on ML [72]. Of the 84 papers, 16 papers reported interpretation methods of ML, constituting 19.05% of all papers (Figure 9). This indicated that explanations for ML outputs were provided in only a few papers. The interpretation methods utilized in the 16 studies included tree-based feature importance (TFI) [73], sensitivity analysis (SA) [74], permutation feature importance (PFI), partial dependence plot (PDP) [75], and Shapley additive explanations (SHAP) [76]. TFI calculates the contribution of each feature and is exclusively applicable to tree models, unlike the other four interpretation methods. Among the 16 studies, 10 studies interpreted ML using TFI, while PFI, PDP, and SHAP were rarely utilized.

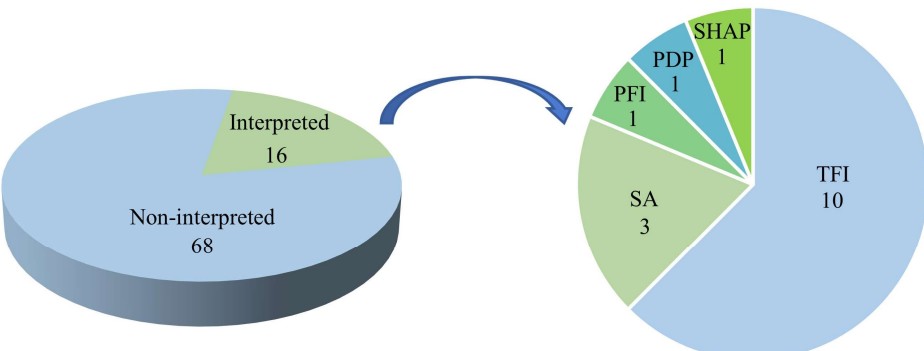

**Figure 9.** The number of studies using each interpretation method for ML.

### 3.2.4. Sample Sources

In the prediction of debris flow occurrence based on ML, the sample data, which serve as the output of the predictor, primarily consist of debris flow events or non-debris flow events in the evaluation unit. The sample data sources in this study were categorized into three groups: searching by data materials (including historical records, official announcements, related websites, etc.), remote sensing interpretation (utilizing high-resolution satellite images and aerial photographs), and field survey; the results are presented in Figure 10. In this study, "searching by data materials" refers to extracting sample data from relevant texts containing information about debris flow events. These texts encompass electronic texts downloaded from online sources and paper texts. Searching by data materials was the most important sample data source in predicting debris flow occurrence based on ML, with 58 studies. Of the 84 studies, 52 used a single sample data source, and

32 utilized a combination of sample data sources. This suggested that a single method was more commonly utilized than a combination of multiple sample data sources in predicting debris flow occurrence based on ML.

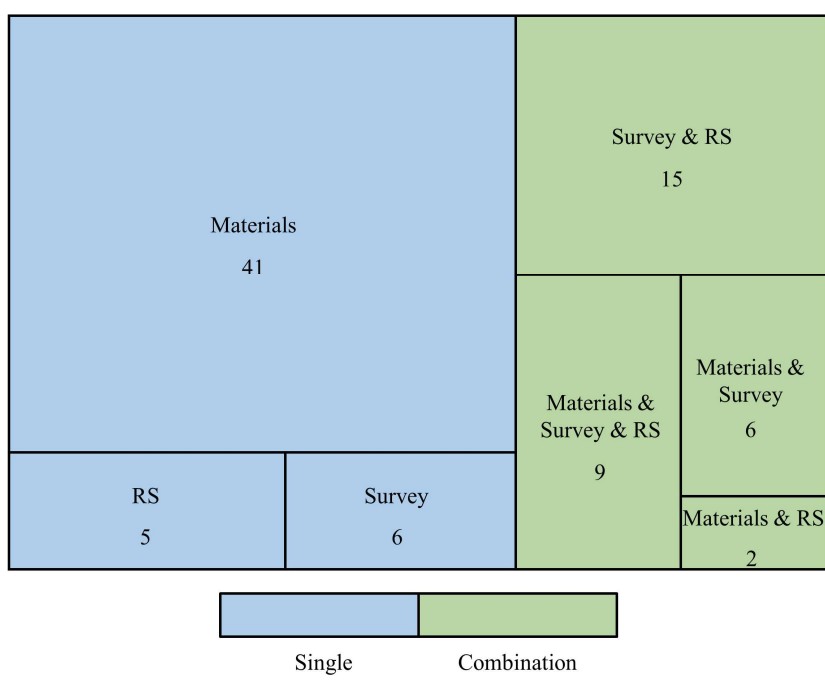

**Figure 10.** The number of studies utilizing each category of sample data sources.

### 3.2.5. Evaluation Units and Candidate Variable Categories

From the 84 papers, the types of evaluation units were classified and summarized, as depicted in Figure 11. The evaluation unit categories were classified into two broad categories, with 49 studies using surface evaluation units and 36 using point evaluation units. Only one paper [77] utilized both categories of evaluation units, and the remaining studies all utilized only one category of evaluation units. The point evaluation units correspond to the grid cell. Regarding the surface evaluation unit, watershed was the most important evaluation unit in predicting debris flow occurrence based on ML, with 21 papers. Only one paper selected village with social property as the evaluation unit, while the remaining studies selected evaluation unit with natural property in predicting debris flow occurrence based on ML.

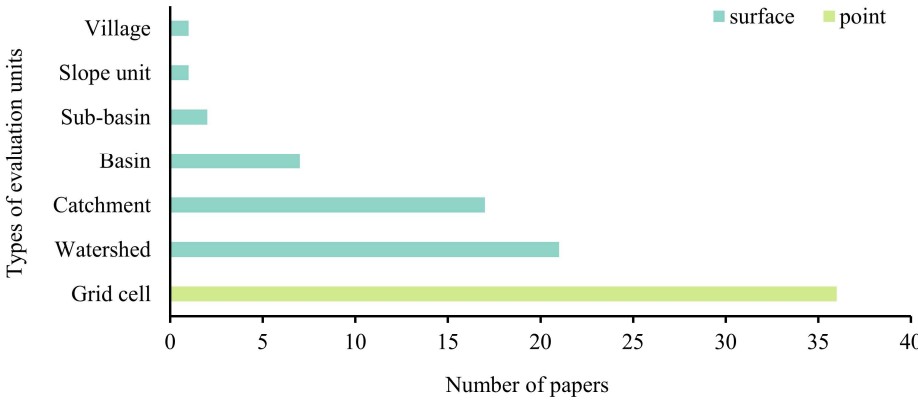

**Figure 11.** The number of studies using each evaluation unit, according to the surface evaluation unit and point evaluation unit reported in the papers.

In the prediction of debris flow occurrence, the depiction of candidate variables is intricately linked to the author's cognition, the research domain, and the chosen evaluation unit. Notably, the representation of identical candidate variables varies across the papers, reflecting the nuances of individual research intentions. Therefore, with reference to relevant studies [24,78–82], the candidate variables from the 84 papers were systematically categorized into 12 groups: topography factors, morphology factors, geomorphology factors, geology factors, meteorology factors, hydrology factors, soil factors, vegetation factors, fire factors, material source factors, human activity factors, and past debris flow characteristic factors (Table 3).

**Table 3.** Classification criteria for candidate variables.

| Category | Description |
|---|---|
| Topography | Factors related to topography, such as slope, curvature, main channel length, etc. |
| Morphology | Factors related to the morphology of the surface evaluation unit, such as area, shape coefficient, perimeter, etc. |
| Geomorphology | Factors related to geomorphic type and evolution, such as landform, hypsometric integra, geomorphic information entropy, etc. |
| Geology | Factors related to geological structure, geological movement, and geological type, such as active fault density, seismic intensity, lithology, etc. |
| Meteorology | Factors related to meteorological factors such as rainfall, temperature, snow cover, etc. |
| Hydrology | Factors related to water flow movement, such as flow accumulation, stream power index, distance to rivers, etc. |
| Soil | Factors related to soil type, property, and thickness, such as soil texture, soil types, soil depth, etc. |
| Vegetation | Factors related to vegetation type and state, such as vegetation coverage index, normalized difference vegetation index, forest density, etc. |
| Fire | Factors related to forest fires, such as fire severity (low, moderate, high), proportion of watersheds burned at high or moderate severity, etc. |
| Material source | Factors related to loosen accumulation of internal solids, such as collapsed areas, landslide areas, debris reserves, etc. |
| Human activity | Factors that directly or indirectly characterize human behavior, such as land use, population density, distance to road, etc. |
| Past debris flow | Factors related to past debris flows in the evaluation unit, such as maximum volume, occurrence frequency, etc. |

The studies using factors from each of the 12 categories were counted from the two aspects of the point evaluation units and surface evaluation units, and the results are shown in Figure 12. Among the 49 papers utilizing the surface evaluation units, all of the 12 categories were used, with the top three being topography factors (45), meteorology factors (40), and morphology factors (35). By contrast, in the 36 studies utilizing point evaluation units, morphology factors, fire factors, material source factors, and characteristics factors of past debris flow were not used; the top three with the highest number of papers reported were topography factor (34), hydrology factor (27), and human activity factor (24). This discrepancy may be due to morphology factors, material source factors, and characteristic factors of past debris flow being associated with surface evaluation units, and researchers were more inclined to utilize the surface evaluation units to predict the occurrence of post-fire debris flow (fire factor utilized) based on ML. In the studies utilizing point evaluation units and surface evaluation units, there was a difference in the ranking of the number of studies using each variable category, with topographic factors being the most popular, with a total of 78 papers. This preference can be attributed to its commendable capacity to represent potential material source and energy, rendering it suitable for predicting diverse debris flow types, and its data accessibility.

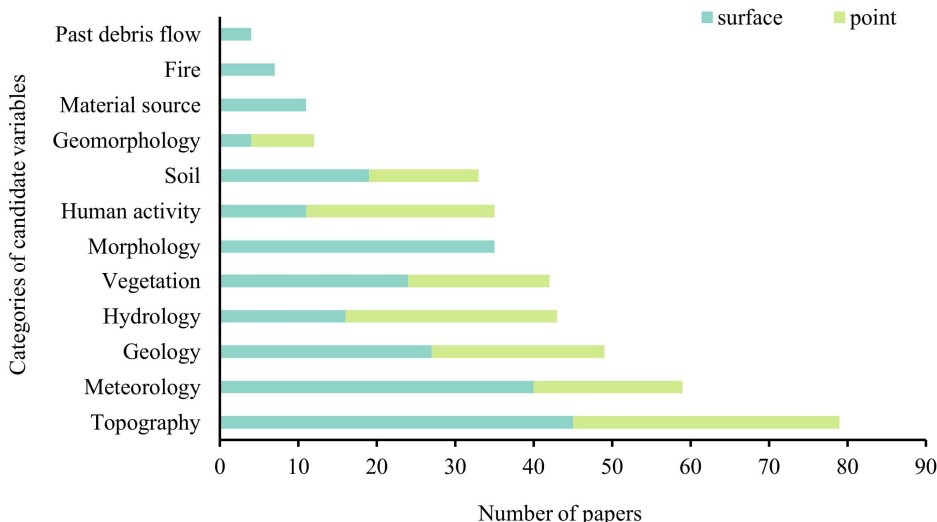

**Figure 12.** The number of studies using each candidate variable category, according to the surface evaluation units and point evaluation units reported in the papers.

### 3.2.6. Validation Techniques and Evaluation Metrics

In the prediction of debris flow occurrence based on ML, it is necessary to select validation techniques and evaluation metrics to evaluate the performance of the model. Among the 84 papers reviewed, two primary validation techniques were utilized: hold-out (61) and cross-validation (23). Notably, hold-out emerged as the most prevalent validation technique in predicting debris flow occurrence based on ML, as illustrated in Figure 13. A total of 21 evaluation metrics were grouped into model fitting metrics (root mean square error (RMSE), mean absolute percentage error (MAPE), R-squared ($R^2$), etc.) and prediction performance metrics (AUROC, overall accuracy (ACC), Kappa coefficient(kappa), etc.). Among the 21 evaluation metrics, 12 evaluation metrics were reported in 2 or more papers, with 3 evaluation metrics reported in over 20 papers, in the following order: AUROC (57), ACC (49), and sensitivity (21), as shown in Figure 12. The prominence of AUROC can be attributed to the fact that ML-based prediction modeling of debris flow occurrence essentially involves classification models, for which AUROC serves as a robust performance measure.

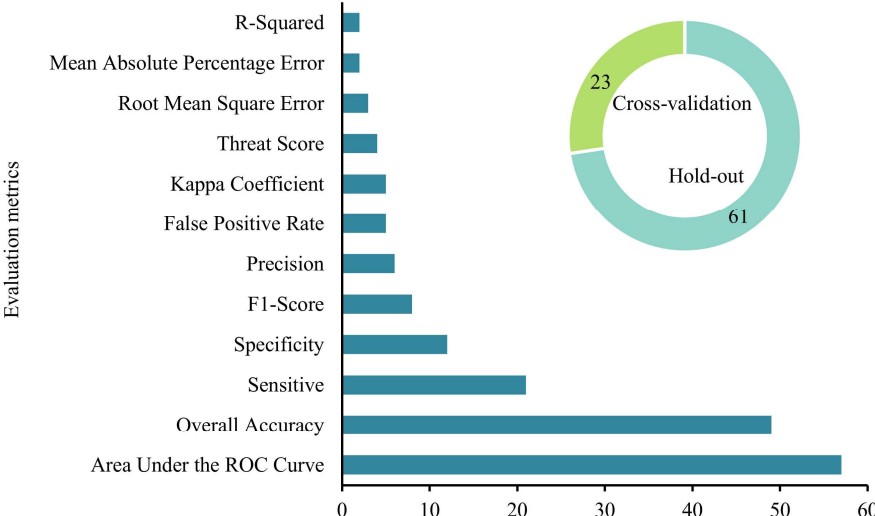

**Figure 13.** The number of studies using each validation technique and evaluation metric (only those evaluation metrics reported in 2 or more papers were included).

### 3.2.7. Prediction Performance

According to Section 3.2.1, the baseline predictors of debris flow occurrence in more than 20 papers were as follows: LR, SVM, BPNN, and RF. Additionally, as indicated in Section 3.2.6, the most utilized evaluation metric was AUROC. Given the sample size, in this study, the prediction performances of the main baseline predictors (LR, SVM, BPNN, and RF) were analyzed based on AUROC.

Figure 14 shows the number of sample data of the four baseline predictors, with the following distribution: LR (38), RF (22), SVM (21), and BPNN (13). The sample data for the four baseline predictors were extracted from their cases using AUROC as the evaluation metric in the 84 studies. The average AUROC for the four baseline predictors was greater than 81%, with the following specific values: RF (0.870), LR (0.859), BPNN (0.845), and SVM (0.816). These results indicated that the four ML models as baseline predictors exhibited good performance in the prediction of debris flow occurrence.

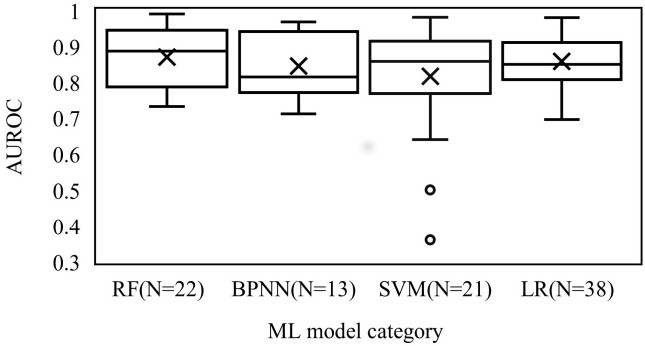

**Figure 14.** AUROC of different baseline predictors. The baseline predictors were logistic regression (LR), support vector machine (SVM), back-propagation neural network (BPNN), and random forest (RF).

Among the 84 studies, different studies utilized different ML models, input variables, and data sets. To compare the prediction performances of the four models, pairs of baseline predictors were selected, and the results are illustrated in Figure 14. The number of sample data of each pairwise comparison was as follows: LR and RF (6), LR and BPNN (3), LR and SVM (5), SVM and RF (10), BPNN and RF (6), BPNN and SVM (10). The sample data of six pairs of baseline predictors were extracted from their cases using AUROC as an evaluation metric in the 84 studies. In Figure 15a, two points are above the 1:1 line, and the remaining points are below. In Figure 15b, the three points are all below, but very close to the 1:1 line. In Figure 15c, only one point is above the 1:1 line, and the remaining points are below. Therefore, on the whole, the prediction performance of LR was worse than that of RF, BPNN, and SVM in the prediction of debris flow occurrence. In Figure 15d, five points are above the 1:1 line, four points are below, and one point is on the line, indicating that the prediction performance of SVM as a baseline predictor was comparable to that of RF in the prediction of debris flow occurrence. In Figure 15e, only one point is above the 1:1 line, and the remaining points are below, indicating that the prediction performance of BPNN as the baseline predictor was worse than RF in the prediction of debris flow occurrence. In Figure 15f, three points are above the 1:1 line, six points are below, and one point is on the line, indicating that the prediction performance of SVM as a baseline predictor was better than that of BPNN in the prediction of debris flow occurrence, but there was no absolute advantage.

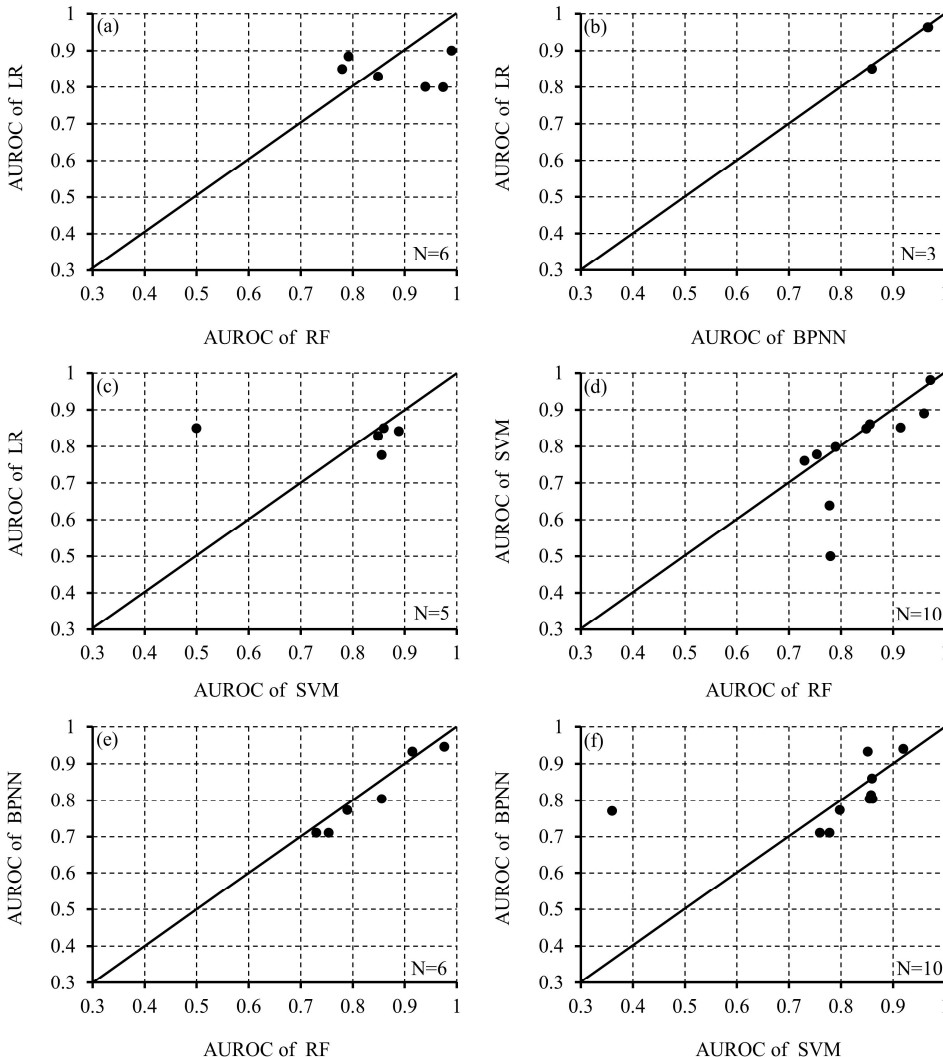

**Figure 15.** Comparison of selected pairs of baseline predictors based on AUROC.

3.2.8. Application Processes

Figure 16 summarizes the processes involved in the prediction of debris flow occurrence based on ML, as extracted from the 84 papers. The processes consisted of three main steps: data preparation, model construction and evaluation, and prediction outcomes. In the data preparation process, first, the evaluation unit for the study area is chosen, such as a watershed, catchment, or grid cell. Then, debris flow sample data are collected as output from one or more sample data sources, including searching by data materials, field survey, and remote sensing interpretation. Next, candidate variables containing geo-environmental information are extracted based on the relevant literature and expert experience, considering the availability and reliability of data (remote sensing data, digital elevation models, thematic maps, etc.). Finally, the raw dataset of prediction of debris flow occurrence is constructed. In the model construction and evaluation process, first, the raw dataset is split into training and testing datasets (few studies split the raw dataset sets into training, validation, and testing datasets), with some studies utilizing only cross-validation to evaluate model performance. Then, suitable ML models (BPNN, SVM, CNN, etc.) are selected as baseline models based on different research purposes and the characteristics of the ML models. Next, certain improvement strategies are implemented, such as feature engineering, hyperparameter tuning, and model coupling to improve the baseline model's prediction performance, or some studies directly utilized the original ML models. Finally, suitable evaluation metrics (AUROC, ACC, RMSE, etc.) are selected to evaluate the perfor-

mance of the model, and the optimal model is selected according to the evaluation results. In the prediction outcome process, the statement of theory assumes that "past events have a great influence on the future" [83] is necessary. The prediction of debris flow occurrence in future situations based on the optimal model is performed, in other words, determining whether debris flow will occur or the possibility of debris flow occurrence (susceptibility assessment or hazard assessment).

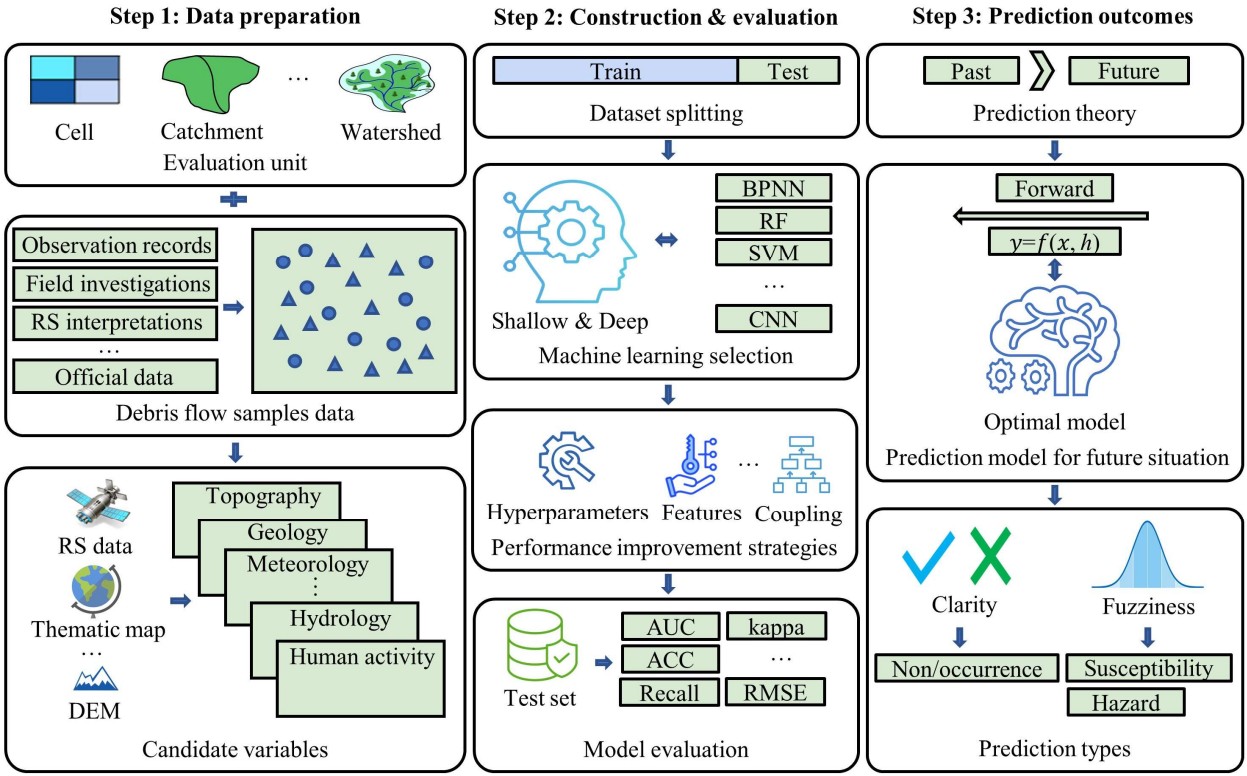

**Figure 16.** Three main steps in predicting debris flow occurrence based on ML.

## 4. Discussion

### 4.1. Challenges and Future Trends

Without a doubt, ML is a promising approach for the prediction of debris flow occurrence, as evidenced by the analysis of 84 papers. However, because of the limited skills of debris flow disaster researchers in ML and the lack of modeling data, there are still shortcomings and challenges in the current research. First, the utilization of new ML techniques in the prediction of debris flow occurrence has an obvious lag. For instance, according to Section 3, reinforcement learning and transfer learning were rarely used, and compared with shallow ML, deep learning was also less commonly utilized. Second, the application of interpretation methods in the prediction of debris flow occurrence based on ML lacked breadth and depth. For example, as indicated in Section 3.2.3, most of the selected papers did not interpret the ML results, and the few that did employ limited categories of interpretation methods to analyze feature importance did not use model visualization or provide post-hoc explanations. To address these problems, we propose the following two general recommendations for future research to seek suitable solutions. The details of the recommendations are as follows:

- ML is evolving rapidly, and the utilization of new ML techniques may revitalize the research of prediction of debris flow occurrence. On the one hand, educating geoscientists on the advantages of utilizing new techniques, such as deep learning, reinforcement learning, and transfer learning, to predict debris flow occurrence. On

the other hand, the integration of domain knowledge of debris flow occurrence with the new techniques should be further explored.

- Comparing various features of explainable frameworks, such as SHAP and local interpretable model-agnostic explanations (LIME) [84], and selecting suitable interpretation methods could improve the transparency and credibility of ML in the prediction of debris flow occurrence. Model visualization and post-hoc explanations should be given more attention to provide insights into the utilization of ML as a predictor of debris flow occurrence. Furthermore, through mechanism-learning coupling methods, such as mechanism cascaded learning, learning-embedded mechanisms, and mechanism-integrated learning, mechanism models and ML models can be combined to improve the physical interpretability for prediction outcomes of debris flow occurrence [85].

*4.2. Uncertainties and Limitations*

The potential uncertainties in the results and limitations of this meta-analysis are outlined below.

- Collection of papers: While a considerable effort was invested in defining the search criteria for ML and debris flow occurrence, we may omit certain papers. Additionally, the scope of this study was confined to papers published in English-language journals. It is worth acknowledging that numerous studies, particularly in regions susceptible to debris flow that are non-English speaking, may have been published in other languages such as Chinese, Japanese, and Portuguese. This language restriction could potentially exclude relevant contributions in languages other than English.
- Prediction performance of ML: Given the sample size, the quantitative analysis of prediction performance was limited to the four most frequently reported ML models (LR, SVM, BPNN, and RF), neglecting potential insights from less-reported models. In addition, variations in the evaluation units, study areas, and types of debris flow were not accounted for, potentially influencing the results of the quantitative analyses.

## 5. Conclusions

In this study, a meta-analysis of the research on the prediction of debris flow occurrence based on ML was conducted by reviewing the relevant papers from the Scopus and Web of Science databases. A summary of this study's content and crucial findings are presented below.

- A total of 84 papers were published from 2006 to 2023, with an overall rising trend, particularly in recent years (2018–2023), suggesting an increasing interest in predicting debris flow occurrence based on ML. Debris flow disasters occur throughout the world, and many countries have carried out research on the prediction of debris flow occurrence based on ML; China has made significant contributions, but more research efforts in African countries should be considered.
- A total of 36 categories of ML models were utilized as baseline predictors for debris flow occurrence. Notably, extreme gradient boosting, gradient tree boosting, convolutional neural network, and multilayer perceptron had strong popularity in predictive modeling of debris flow occurrence. Additionally, LR and RF emerged as the most popular ML models in predicting debris flow occurrence.
- In the prediction of debris flow occurrence based on ML, a variety of prediction performance improvement strategies, including feature engineering, model comparison, hyperparameter tuning, model coupling, and structure optimization, were widely utilized. Among these strategies, feature engineering and model comparison emerged as the most common strategies; the most common approach for model improvement in predicting debris flow occurrence based on ML involved the utilization of one or two of these strategies, while fewer studies utilized three or four of these strategies.
- In the prediction of debris flow occurrence based on ML, few papers provided interpretation methods of ML; searching by data materials emerged as the most crucial debris flow sample data source. There was a difference in the ranking of the number of

studies using each candidate variable category between the studies utilizing point evaluation units and those using surface evaluation units, but the number of topographic factors was the highest. Two validation techniques, hold-out and cross-validation, were utilized. AUROC was the most frequently reported evaluation metric, followed by ACC, sensitivity, and specificity.

- The four ML models (RF, LR, BPNN, and SVM) used as baseline predictors exhibited good prediction performance in the prediction of debris flow occurrence. LR's prediction performance for debris flow occurrence was inferior to RF, BPNN, and SVM; SVM was comparable to RF, and all were superior to BPNN.
- The process of predicting debris flow occurrence based on ML consisted of three main steps: data preparation, model construction and evaluation, and prediction outcomes.
- Future work on the prediction of debris flow occurrence based on ML can focus on two aspects: utilizing new ML techniques, and enhancing the interpretability of the ML models.

**Author Contributions:** Conceptualization, L.Y. and Y.G.; Supervision, Y.G.; Formal Analysis, L.Y. and Y.G.; Data Collection, L.Y.; Visualization, L.Y., B.C. and Y.W.; Writing—Original Draft Preparation L.Y., B.C. and R.F.; Writing—Review and Editing, L.Y., Y.G., B.C., Y.W. and R.F. All authors have read and agreed to the published version of the manuscript.

**Funding:** This research was supported by the Second Tibetan Plateau Scientific Expedition and Research Program (STEP) (Grant No. 2019QZKK0902).

**Data Availability Statement:** The data presented in this study can be made available upon request from the authors. The data are not publicly available due to privacy restrictions.

**Conflicts of Interest:** The authors declare no conflicts of interest.

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
