# Peer review of "Machine-Learning-Based Prediction Modeling for Debris Flow Occurrence: A Meta-Analysis"

_water, doi:10.3390/w16070923_

Round 1

Reviewer 1 Report

Comments and Suggestions for Authors

The manuscript “Machine Learning-Based Prediction Modeling for debris flow Occurrence: A Meta-Analysis” analyzes the use of ML to predict debris flows in the literatures since 2000. The review is comprehensive and will be a good addition to the literature.

However, there needs to be significant work to make this work publishable, especially in a higher-tier journal such as MDPI-Water.

The work appears to be complete and have used established methods. The issues are more with presentation and communication.

The paper is difficult to read. In some cases this is because of the use of non-standard English. The manuscript would benefit from editing by an native English speaker familiar with both ML and debris flow prediction.

The graphics are inconsistent and difficult to interpret. The authors should pick one style and stick with it. I like the Ven-diagram approach used in Fig 6. The blocks in Fig 8 and 9 are Ok. The circular bar graph in Fig 7 is very difficult to interpret.

I’m surprised at the focus on AURC as an error metric. While this is the standard in signal processing literature, it is less common in hydrology literature. I don’t understand why it was used as the basis for comparison. I was also surprised that AURC was used in 60 of the 84 papers.

Many of the paragraphs the present the data on specific issues, (e.g., L182-200 and other similar) would be better as tables, with just the highlights discussed. This is true for almost every section.

I would like a few more details on how the papers returned from the search, 410, were reduced to 84. The manuscript only states “after meticulous review”, which is not a description of the methods used.

I provided a markup of the manuscript with a number of things noted. I did not attempt to edit the English, though I did markup a few instances.

L518-579 The majority of these statements did not come from your analysis, or if it did, you didn’t present the data. For example, you state that “low interpretavility” is a major concern, but you never state what percentage of papers make that claim. Again, most of these claims are not based on data presented from you study. They seem more the standard laundry list that could have been written without the study.

Specific comments

Abstract: Add some more numbers/stats to summarize the study (see markup for specific issues)

L42-44 English is difficult to follow

L119 Figure 1. Just list the search terms. Don’t repeat the “ANDS” or “ORS”, maybe put in a block or something and state which is used. The figure is fuzzy and difficult to read.

L121 Figure 2 There are a number of issues with this figure (see markup). For example, the numbers don’t work (see markup)

L130 Why does AURC have its own field in the database? Non of the other metrics do.

L139-160 1st P of Section 3 needs to be re-written. Confusing and difficult to follow.

L215-219  I don’t follow this discussion. Problems with English – need to be re-written.

L220-234 You define “categories” several times with several different numbers. I assume these are subcategories. Need the English to be clear and precise. Currently it is confusing and difficult to follow. (see markup)

L220-234 (and elsewhere) always need to spell out acronyms on 1st use.

L239-25 Need to describe how you computed the “relative annual number” and what it means.

L252 Caption talks about “slashes” in the figure – there are none.

L360 Section 3.2.3 What does “searching by data material” mean? Need to define terms and describe methods.

L397 Table 3 Make the 1st column wider so Category names don’t wrap.

L450 What is greater than 20? It isn’t clear.

L482 Figure 13. These numbers add up to 84 (total number of papers), but only 60 papers had AURC as a metric. Where did these data come from?

L486 Figure 14. I’m surprised that this many papers had both AURC and the other associated metrics so you could make these plots. This doesn’t seem to follow Fig 12. Maybe need to do better with Fig 12 and show which papers have more than one metric.

L515 Figure 15 Fuzzy difficult to read. I’m not sure how useful this figure is.

L518-579 The majority of these statements did not come from your analysis, or if it did, you didn’t present the data. For example, you state that “low interpretavility” is a major concern, but you never state what percentage of papers make that claim. Again, most of these claims are not based on data presented from you study. They seem more the standard laundry list that could have been written without the study.

L539-547 While I agree strongly with this statement (ML is a blackbox), it isn’t supported by data you report from your study.

Comments on the Quality of English Language

While English grammar and spelling are good, the sentence structure, organization, and language use makes the manuscript very difficult to follow.

Reviewer 2 Report

Comments and Suggestions for Authors

This paper focuses on the modeling of debris flow occurrence prediction based on machine learning. It systematically summarizes the trends, distribution of research institutions, model algorithms used, and data sources for modeling in related studies. The analytical methodology of the paper is scientific and rigorous. The conclusions drawn from the research have important reference value for the field, providing readers with a comprehensive understanding of the current research status and development trends. However, there are some obvious shortcomings in the paper that need to be improved, as follows:

1.         There are issues with the numbering of section titles in the paper. The title at line 123 should be numbered as "2.2."

2.         There are problems with the section titles. The titles at line 160 and line 209 are duplicated. The latter should be the characteristics of the research method.

3.         In Figure 14, it is suggested to combine the six images into one, using different colors or shapes to represent different models. This will enrich the information in the image and make it easier to compare the predictive performance of different models.

4.         The images in the paper are generally distorted, possibly due to compression. The author needs to improve the clarity of the images.

5.         In Figure 4, it is suggested to use different colors to represent the number of related research papers from each country, in order to enhance the readability of the image. For example, using progressively deeper colors to indicate the increasing number of papers.

6. The related studies using the rainfall threshold method also include: http://dx.doi.org/10.1007/s10064-023-03068-9.

Round 2

Reviewer 1 Report

Comments and Suggestions for Authors

The authors have addressed my comments.